# Development of a Method for Scaffold-Free Elastic Cartilage Creation

**DOI:** 10.3390/ijms21228496

**Published:** 2020-11-11

**Authors:** Masahiro Enomura, Soichiro Murata, Yuri Terado, Maiko Tanaka, Shinji Kobayashi, Takayoshi Oba, Shintaro Kagimoto, Yuichiro Yabuki, Kenichi Morita, Toshimasa Uemura, Jiro Maegawa, Hideki Taniguchi

**Affiliations:** 1Department of Regenerative Medicine, Graduate School of Medicine, Yokohama City University, 3-9 Fukuura, Kanazawa-ku, Yokohama 236-0004, Japan; t136012c@yokohama-cu.ac.jp (M.E.); terado@yokohama-cu.ac.jp (Y.T.); mytanaka@yokohama-cu.ac.jp (M.T.); 2Division of Regenerative Medicine, Center for Stem Cell Biology and Regenerative Medicine, The Institute of Medical Science, the University of Tokyo, 4-6-1 Shirokanedai, Minato-ku, Tokyo 108-8639, Japan; 3Department of Plastic and Reconstructive Surgery, Kanagawa Children’s Medical Center, 2-138-4 Mutsukawa, Minami-ku, Yokohama 232-8555, Kanagawa, Japan; skobayashi@kcmc.jp; 4Department of Orthopaedic Surgery, Graduate School of Medicine, Yokohama City University, 3-9 Fukuura, Kanazawa-ku, Yokohama 236-0004, Kanagawa, Japan; t186017g@yokohama-cu.ac.jp; 5Department of Plastic and Reconstructive Surgery, Yokohama City University Hospital, 3-9 Fukuura, Kanazawa-ku, Yokohama 236-0004, Kanagawa, Japan; kagishin@hotmail.co.jp (S.K.); yu1rou@hotmail.co.jp (Y.Y.); jrmaegawa@gmail.com (J.M.); 6Cell Culture Research Center, JTEC COOPERATION, Ibaraki 567-0086, Osaka, Japan; kenichi.morita@j-tec.co.jp (K.M.); uemura@prec.eng.osaka-u.ac.jp (T.U.); 7Graduate School of Engineering, Osaka University, Suita 565-0871, Osaka, Japan

**Keywords:** elastic cartilage, scaffold-free, three-dimensional rotating suspension culture, chondrocyte progenitor cells

## Abstract

Microtia is a congenital aplasia of the auricular cartilage. Conventionally, autologous costal cartilage grafts are collected and shaped for transplantation. However, in this method, excessive invasion occurs due to limitations in the costal cartilage collection. Due to deformation over time after transplantation of the shaped graft, problems with long-term morphological maintenance exist. Additionally, the lack of elasticity with costal cartilage grafts is worth mentioning, as costal cartilage is a type of hyaline cartilage. Medical plastic materials have been transplanted as alternatives to costal cartilage, but transplant rejection and deformation over time are inevitable. It is imperative to create tissues for transplantation using cells of biological origin. Hence, cartilage tissues were developed using a biodegradable scaffold material. However, such materials suffer from transplant rejection and biodegradation, causing the transplanted cartilage tissue to deform due to a lack of elasticity. To address this problem, we established a method for creating elastic cartilage tissue for transplantation with autologous cells without using scaffold materials. Chondrocyte progenitor cells were collected from perichondrial tissue of the ear cartilage. By using a multilayer culture and a three-dimensional rotating suspension culture vessel system, we succeeded in creating scaffold-free elastic cartilage from cartilage progenitor cells.

## 1. Introduction

Microtia is a congenital aplasia of the auricular cartilage. Conventionally, autologous costal cartilage grafts are collected and shaped for transplantation for microtia treatment. However, this method has three problems. First, there is an excessive invasion associated with the collection of costal cartilage. Second, the collection of the autologous costal cartilage is limited. Lastly, long-term morphological maintenance is required because of the deformation of transplanted grafts over time [1]. Additionally, since the costal cartilage grafts used are composed of hyaline cartilage, the lack of elasticity is worth mentioning for the treatment of microtia. Medical plastic materials have been transplanted as alternatives to costal cartilage; however, unfortunately transplant rejection and deformation over time occur using this method. There is a need to create a tissue for transplantation using autologous elastic cartilage cells [2,3].

In 1997, the generation of engineered cartilage with a human auricular shape in a nude mouse model was reported [4]. Other studies reported the creation of engineered human ears shaped in vitro and in vivo using scaffolds [5,6,7,8,9].

Attempts have been made to create cartilage tissue for transplantation using a biodegradable scaffold material to address this problem. However, scaffolding materials cannot avoid transplant rejection. Biodegradation of the scaffolding material after transplantation causes the transplanted cartilage tissue to deform, which leads to a lack of shear stress [10]. Consequently, a method needs to be established for scaffold-free elastic cartilage tissue creation for transplantation using autologous cells. Nevertheless, elastic cartilage tissue is known to function by arranging chondrocytes three-dimensionally, not two-dimensionally [11]. The conventional scaffolding-free cartilage tissue creation method uses tissue measuring 3 cm^2^. The implant used in a previous study on tissue thickness did not reach the auricle. Additionally, it did not mention the shear stress, rather it only stated that protocol modification is necessary because elastin and the elastic fibers are not synthesized in vitro [12]. Further, the required size of the auricular cartilage for microtia treatment is 2–3 cm in length [2,7], meaning about 8 cm^3^ of cartilage tissue is required.

This study aims to establish a method for the creation of scaffold-free elastic cartilage tissue with hardness and thickness levels comparable with auricular cartilage.

## 2. Results

### 2.1. Isolation of Cartilage Progenitor Cells

The auricular cartilage tissue of a living body can be confirmed by Alcian blue (AB) and elastica–van Gieson (EVG) staining. Further, whether or not the periphery of the cartilage tissue expressing collagen type Ⅱ (Col2) has a layered structure covered with collagen type Ⅰ (Col1)-positive perichondrium tissue can be confirmed. Our previous study revealed that chondrocyte progenitor cells are present in perichondrial tissue [2,3]. This perichondrial tissue was isolated and collected, and the chondrocyte progenitor cells present inside were cultured (Figure 1). Chondrocyte progenitor cells can be stably cultured in a subculture, and depending on the number of passages there may be no difference in the observed cell morphology (Figure 2A). However, when the cartilage function was evaluated using melanoma-inhibiting activity (MIA) measurement, there was a difference in the amount of MIA between P3-6 and P10-13 cells (Figure 2C). Previous studies have shown that the dedifferentiation of chondrocyte progenitor cells occurs during multiple passages over time [11,13,14]. Thus, improving the function of chondrocyte progenitor cells to create cartilage tissue that retains its elasticity is important [13,15,16].

### 2.2. Generation of Cultured Cartilage for Multilayer and Rotating Cultures

Multilayer [17] and three-dimensional rotation cultures are known to enhance the function of cartilage progenitor cells [12,18,19,20,21]. Chondrocyte progenitor cells were cultured by multiple layers to form a multilayer sheet (Figure 1B). By using a multilayer culture, the chondrocyte progenitor cells secreted more MIA than the monolayer culture (Figure 2C). Moreover, it was confirmed that the content of hyaluronic acid (HA) also increased as the number of multilayers increased (Figure 2D). MIA and hyaluronic acid are the typical markers of the extra cellular matrix (ECM) content of chondrocyte progenitor cells. By using a rotation culture using the rotating wall vessel (RWV) for 3 weeks (Figure 3A), the chondrocyte progenitor cells formed a three-dimensional tissue with increased thickness (Figure 3B). Moreover, the content of MIA was significantly improved compared with that of the 2D culture (Figure 3C). The elastic force was measured via the shear stress, which was increased in RWV culture for the cultured elastic cartilage (Figure 3D). Thus, the three-dimensional elastic cartilage tissue was created using chondrocyte progenitor cells combined with multilayer and rotating cultures (Figure 3A).

### 2.3. Maturation of Transplanting Cultured Cartilage

The cultured elastic cartilage tissue created using the RWV was immunohistochemically positive only for Col1 but negative when stained in AB or EVG (Figure 4A). When the physical elastic force was measured via the shear stress, a difference was found when compared with the primary auricular cartilage (Figure 4B). Therefore, it was shown that the cultured elastic cartilage tissue was immature. For further maturation, the cultured cartilage tissue was transplanted subcutaneously on the back of a mouse for 2 months. The cartilage tissue after transplantation had a layered structure covered with Col2-positive elastic cartilage tissue and Col1-positive perichondrium tissue. AB and EVG staining also showed maturation of the cultured elastic cartilage tissue (Figure 4A). Mechanical measurements also showed that the transplanted elastic cartilage tissue had a similar elastic force as that of the primary auricular cartilage (Figure 4B).

### 2.4. Long-Term Changes of the Cartilage Tissue after Transplantation

The transplanted elastic cartilage tissue showed significant shrinkage 2 weeks after transplantation, but after that the tissue became stable and there was no significant difference between 1 and 2 months after transplantation. Thus, it was confirmed that the transplanted elastic cartilage tissue contracts during the maturation process after transplantation (Appendix A
Section 1). After maturation, these elastic cartilage tissues remained unchanged in shape for 2 months. This seems to be due to the maintenance of elastic cartilage tissue by the chondrocyte progenitor cells existing in the perichondrium structure. In addition, we found no immunogenicity of the transplanted elastic cartilage tissue, as shown in Appendix A
Section 2.

## 3. Discussion

We succeeded in reconstructing a three-dimensional elastic cartilage tissue by culturing chondrocyte progenitor cells from perichondrium tissue derived from a human specimen. As shown in the Appendix A, the cells used were not confirmed to have immunogenicity, and early clinical application is expected. Moreover, it is known that mature chondrocytes fail to maintain the phenotype after four or more passages and dedifferentiation [11,13,14]. By enhancing the function of cartilage progenitor cells via the layering and rotation cultures conducted at this time, a three-dimensional cartilage tissue measuring 1000 mm^3^ was formed and we succeeded in reproducing a hardness similar to that of human auricular cartilage (Figure 4B). Previous studies have failed to create thick cartilage tissue. The cartilage tissue created in this study mainly comprised collagen type 2, which is a major component of elastic cartilage tissue. Previous studies reported that human auricular cartilage tissue has a morphology in which the periphery of the elastic cartilage tissue is surrounded by perichondrium tissue. Chondrocyte progenitor cells present in perichondrial tissue maintain the morphology of elastic cartilage tissue [2,3]. The morphology of regenerated cartilage tissue is similar to that of primary auricular cartilage tissue. Regenerated cartilage tissue is similar to the histology of the auricular cartilage tissue covered by the perichondrial tissue, shown as collagen type 1. It is expected that the cartilage progenitor cells present in the perichondrium will maintain the long-term morphology of the regenerated elastic cartilage tissue. The transplanted cartilage tissue contracted mildly during 1 week after transplantation; this tissue was stably maintained for 2 months thereafter, and long-term morphological maintenance is expected. We revealed these transplanted cartilage tissues to have no immunogenicity (Appendix A
Section 2).

The reconstructed cartilage tissue using a scaffold material caused a reduction due to biodegradation of the scaffold material [12]. In addition, the reconstructed cartilage tissue was degraded 2 to 4 years after transplantation by inflammatory reaction [22,23]. On the other hand, our cartilage tissue was free of scaffold material and surrounded by perichondrium-like tissue containing cartilage progenitor cells, so long-term size maintenance would be expected.

Unlike the costal cartilage transplantation that has been conducted for the conventional treatment of microtia, cells recovered from the auricular cartilage can be used for minimally invasive treatment. In addition, even if the number of perichondrial cells is large, three-dimensional regenerated cartilage can be created by applying this method.

As a subject of this method, long-term culturing is required in an RWV involving layering (3 week cultivation by layering and 3 week cultivation by RWV, for a total of 6 weeks). The generation of the three-dimensional structure required long-term cultivation, which is expected to be shortened by examining the culture conditions. Although 10 cm^3^ of cartilage tissue is required for the treatment of microtia, our method without scaffolds is limited to 1 cm^3^ of cartilage tissue (Figure 3B). As an improvement of this problem, it is expected that transplanting a plurality of cultured cartilage tissues and connecting them will create a large number of regenerated cartilage tissues in vivo. 

## 4. Materials and Methods 

### 4.1. Study Approval

We bred and maintained the mice according to our institutional guidelines for the care and use of laboratory animals. All animal studies were carried out following approval from the Institutional Animal Care Use Committee of Yokohama City University (approval no. F-A-20-020). The samples were used according to the permission of the moral board (Yokohama City University Graduate School of Medicine Hospital approval no. B130905006).

### 4.2. Animals

Female NOD/Scid mice aged 10 ± 2 weeks were purchased from Charles River Laboratories Japan, Inc. (Charles River Laboratories, Yokohama, Japan).

### 4.3. Isolation of Chondrocyte Progenitor Cells and Culture Method

We obtained elastic cartilage samples from microtia patients following the approved guidelines set by the ethical committee at Yokohama City University Graduate School of Medicine Hospital (approval no. B130905006). We stripped off the adipose tissue and microscopically separated the cartilage into three layers: the chondrium layer, interlayer, and perichondrium layer. The perichondrial sample was minced with scissors until almost no lumps of the cartilage matrix were found and the perichondrial cells were separated by shaking in a 0.2% collagenase solution (Worthington, Lakewood, NJ, USA) at 37 °C and 600 rpm for 2 h. The resulting suspension was filtered through a 40 µm cell strainer and centrifuged at 4 °C and 1500 rpm for 5 min to collect perichondrocytes. Having been suspended in growth medium and seeded on an uncoated 35 mm dish, the final concentration contained 10% fetal bovine serum (Biowest, Riverside, MO, US), Dulbecco’s Modified Eagle’s Medium (D-MEM)/Nutrient Mixture F-12 Ham (1:1) (Sigma-Aldrich, St. Louis, MO, US), and stabilized antibiotic–antimycotic solution (100×) (Sigma-Aldrich, St. Louis, MO, US). When the perichondrocytes adhered and became confluent, the cells were collected using 0.05% trypsin–EDTA (1×) (Thermo Fisher Scientific, Waltham, MA, US) and expanded in a 225 cm^2^ flask (*n* = 4).

### 4.4. Multilayer Culture 

Chondrocyte progenitor cells were seeded with 1.5 x 10^5^ cells on uncoated 100 mm dishes cultured in a growth medium for 1 day. The next day, after confirming adhesion, the medium was exchanged with a differentiation induction medium and cultured for 6 days; the final concentration contained D-MEM/F12 Ham (1:1) (Sigma-Aldrich, St. Louis, MO, US), stabilized antibiotic–antimycotic solution (100×) (Sigma-Aldrich, St. Louis, MO, US), 0.2 mM ascorbic acid 2-phosphate (AA2P) (Sigma-Aldrich, St. Louis, MO, US), 10^−7^ M dexamethasone (Sigma-Aldrich, St. Louis, MO, US), 1× ITS-X (Gibco™), and 5 ng/mL IGF (Sigma-Aldrich, St. Louis, MO, US). After removing the supernatant on the seventh day, the cells were washed with a wash buffer and then seeded with 1.5 × 10^6^ perichondrocytes; the final concentration contained D-MEM/F12 Ham (1:1) (Sigma-Aldrich, St. Louis, MO, US) and stabilized antibiotic–antimycotic solution (100×) (Sigma-Aldrich, St. Louis, MO, US). The next day, after confirming adhesion, the medium was replaced with a differentiation induction medium and cultured for 6 days. This was repeated once more to prepare a three-layer cell sheet; 2.7 × 10^6^ cells of perichondrocytes were used for each cultured cartilage sample.

### 4.5. RWV Culture

As the rotary culture device, a three-dimensional culture system (Rotary Cell Culture System) manufactured by Synthecon, Inc (Synthecon, Houston, TX, USA) was used. After removing the supernatant, the cells were washed with a washing buffer, then the cell sheet was gently collected with a cell scraper. A six-dish cell sheet was placed in a 50 mL disposable culture vessel (Synthecon) containing a differentiation induction medium and allowed to stand for 1 day, with a final concentration of D-MEM (High Glucose) with L-glutamine and phenol red (FUJIFILM Wako Pure Chemical Corporation, Osaka, Japan), stabilized antibiotic–antimycotic solution (100x) (Sigma-Aldrich, St. Louis, MO, US), 50 ug/mL AA2P (Sigma-Aldrich, St. Louis, MO, US), 40 μg/mL L-proline (Sigma-Aldrich, St. Louis, MO, USA), 10^−7^ M dexamethasone (Sigma-Aldrich, St. Louis, MO, US), 2× ITS- X (Gibco™), and 10 ng/mL TGFβ3 (Sigma-Aldrich, St. Louis, MO, US). The next day, the rotation was started and the cells were cultured for 6 days. The medium was replaced on the 7th and 14th days, and the cells were cultivated for a total of 3 weeks.

### 4.6. Cultured Elastic Cartilage Tissue Transplantation to NOD/SCID Mice

After inhaled anesthetization, shaving of hair around the transplant site, and application of 70% ethanol for sanitation, we made an incision with scissors, inserted the tip of the scissors through the cut, and create a space for the cultured cartilage to move while peeling back the subcutaneous and peritoneal layers. Then, the cultured elastic cartilage tissue was inserted into the created space with ring tweezers while picking up the cut end skin. To record the post-transplant progress, the major and minor axes of the transplanted cartilage were recorded. Forty-one mice were used in these experiments.

### 4.7. Measurement of Transplanted Elastic Cartilage Tissue and Harvest Method

The mice were sacrificed using the laboratory facility’s method. The major axis and minor axis of the transplanted cartilage were measured with calipers and pictures were taken from horizontal and vertical directions. At sampling, the skin around the transplant site was pinched with tweezers and incised to surround the graft. While holding the cultured cartilage with ring tweezers, we removed it from the peritoneum subcutaneously or by using scissors. Excessive tissue (mouse subcutaneous tissue, fat) attached to the cultured cartilage was removed.

### 4.8. Histochemistry and Immunohistochemistry

The following primary antibodies were used in this study: human collagen type I (Col1) rabbit immunoglobulin poly IgG (1:200) (ACRIS, SAN, CA, USA), collagen type II (Col2) clone 6B3 (1:200) (Merck, Darmstadt, HE, Germany). The following second antibodies were used: Alexa Fluor 488 goat anti-rabbit IgG1 (1:500) (Invitrogen: Thermo Fisher Scientific, Waltham, MA, USA), Alexa Fluor 555 goat anti-mouse IgG1 (1:500) (Invitrogen). The nuclei were counterstained with 4′,6-diamidino-2-phenylindole (DAPI).

### 4.9. MIA ELISA 

For measurement of MIA, the DuoSet capture, detection, and standard kit was used (R&D Systems, Minneapolis, MN, USA). We dispensed 100 μL each of blank, standard, and sample into a 96-well plate. After adding 100 μL each of capture reagent, the wells were left overnight. Next, they were washed three times with PBS-T 200 μL/well, Block Ace (DC Pharma, Blankenberge, West Flanders, Belium) was added at 300 μL/well at room temperature (RT) 1 h, then they were washed three times with PBS-T at 200 μL/well. The blank, standard, and samples were added into a 96-well plate at 100 μL/well (RT 2 h) and washed three times with PBS-T at 200 μL/well. Then, 100 μL/well of detection reagent was added (RT 2 h) and the wells were washed three times with PBS-T at 200 μL/well. A certificate analysis working concentration was added at 100 μL/well (RT 20 min), then the wells were washed three times with PBS-T at 200 μL/well. TMB solution was added at 50 μL/well (RT) and the color development time was detected by adding HCL at 50 μL/well. The MIA levels (450 nm) and reference levels were measured via the absorbance with a plate reader (540 or 570 nm).

### 4.10. Glycosaminoglycan Assay

For measurement of glycosaminoglycan, we dispensed 100 μL each of standard, sample, and blank into a 1.5 mL tube. After adding 1 mL each of Blyscan dye reagent, we stirred them with a shaker (RT 30 min) and centrifuged them (4 °C, 12,000 rpm, 10 min). Then, 500 μL of dissociation reagent was added, lightly vortexed, stirred with shaker (RT 10 min), centrifuged (4 °C, 12,000 rpm, 5 min), then 200 μL/well of the supernatant was added to a 96-well plate for ELISA and shading. The glycosaminoglycan levels were measured via the absorbance with a plate reader (656 nm). 

### 4.11. Measurement of Shear Stress

The tabletop tester (Shimadzu Corporation EZ-Test EZ-SX jig S346-57829-02) was used to measure shear stress. The indentation at 3 mm/min and the elastic modulus (MPa) at 0.2–0.6 mm were obtained. The indentation of the vitro sample was 2 mm, and that of the in vivo sample was 1 mm.

### 4.12. Statistics

All data are expressed as the mean ± standard deviation. Significant data were examined using the Holm–Sidak multiple comparisons. A *p*-value of <0.05 was considered significant.

## Figures and Tables

**Figure 1 ijms-21-08496-f001:**
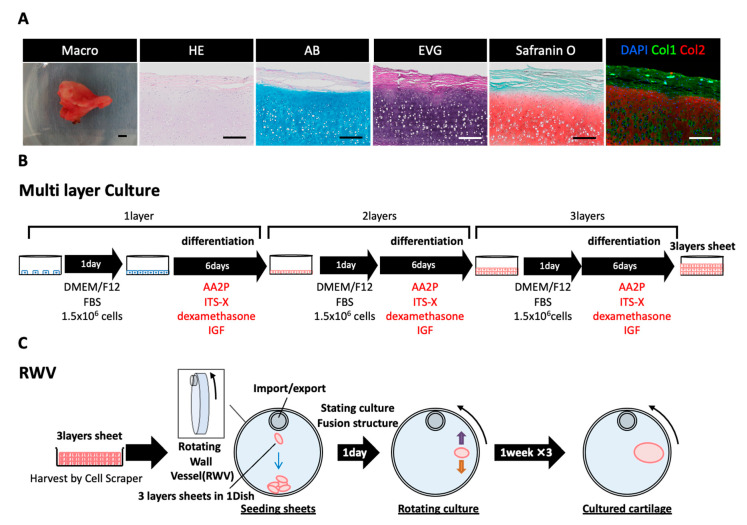
Regenerated cartilage culture method. (**A**) Macro images of human auricular cartilage tissue, along with hematoxylin and eosin (H&E), Alcian blue (AB), elastica–van Gieson (EVG), safranin O, and collagen type I (Col1) and type II (Col2) immunohistological staining. Scale Bars: 200 μm. (**B**) First, 1.5 × 10^6^ chondrocyte progenitor cells collected from the auricular cartilage tissue were seeded, cultured in 1 day DMEM/F12 FBS, and then cultured in a differentiation induction medium for 6 days. After culturing, 1.5 × 10^6^ chondrocyte progenitor cells were seeded and the same operation was repeated twice on a multilayer sheet. (**C**) The multilayer sheet was collected with a scraper, placed in a three-dimensional rotating wall vessel (RWV), and allowed to stand overnight. The RWV rotation culture was conducted so that the multilayer sheet floated.

**Figure 2 ijms-21-08496-f002:**
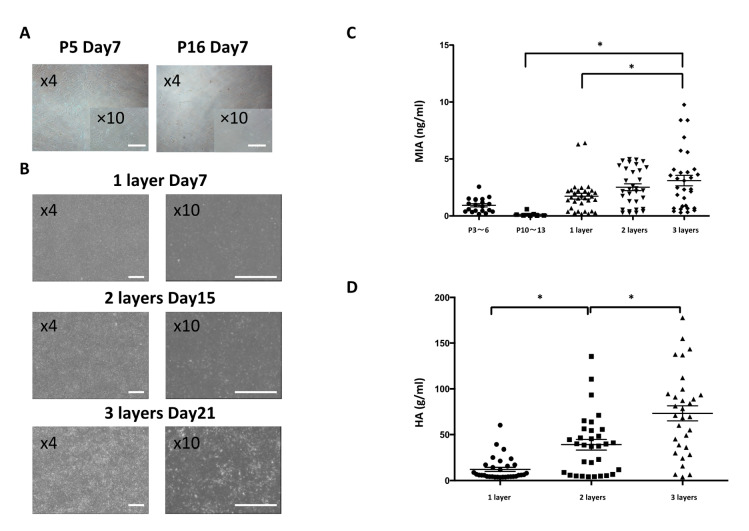
Chondrocyte progenitor cell multilayer culture. (**A**) Culture images of chondrocyte progenitor cells passaged 5 and 16 times. Scale Bars: 500 μm. (**B**) Culture images of chondrocyte progenitor cells in multilayer culture numbers 1 to 3. (**C**) Measurement of melanoma-inhibiting activity (MIA) during subculture and layered culture of chondrocyte progenitor cells. Holm–Sidak multiple comparisons test: * *p* < 0.01 vs. 2D culture and layer number (*n* = 7–26). (**D**) Measurement of hyaluronic acid secretion by layered culture. Holm–Sidak multiple comparisons test: * *p* < 0.01 vs. layer number (*n* = 26)

**Figure 3 ijms-21-08496-f003:**
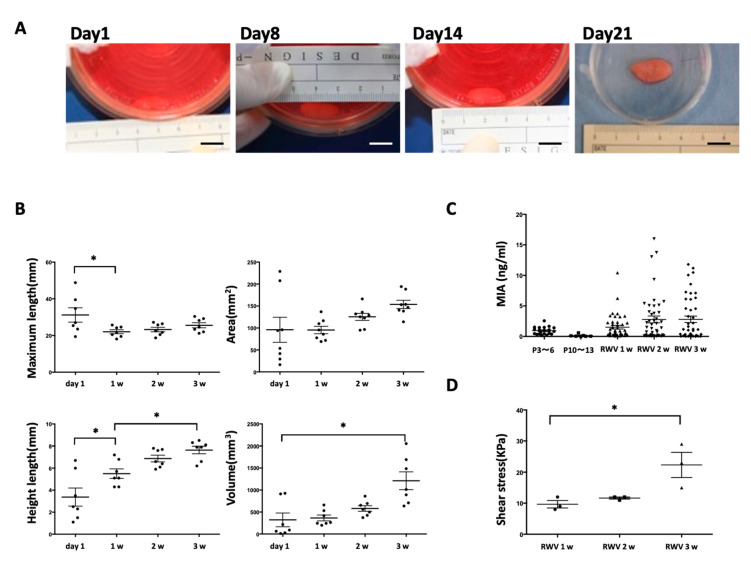
Three-dimensional multilayer cultured elastic cartilage. (**A**) Multilayer sheets cultured in the RWV at 1 day and 1, 2 and 3 weeks. Scale Bars: 1 cm. (**B**) RWV quantitative analysis using Holm–Sidak multiple comparisons test: * *p* < 0.01 vs. RWV culture (*n* = 20–26). (**C**) Measurement of MIA secretion in 2D culture and RWV culture for chondrocyte progenitor cells. (*n* = 7–45). (**D**) Measurement of shear stress in RWV culture at 1, 2, and 3 weeks. Holm–Sidak multiple comparisons test: * *p* < 0.01 vs. RWV culture 3w (*n* = 3).

**Figure 4 ijms-21-08496-f004:**
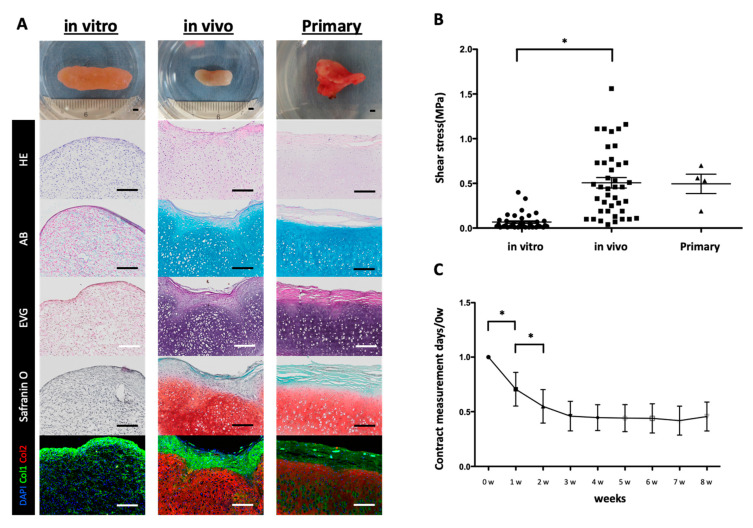
Transplanted elastic cartilage tissue showing maturation. (**A**) Macroscopic images, along with hematoxylin and eosin (HE), Alcian blue (AB), elastica–van Gieson (EVG), safranin O, DAPI, collagen type I (Col1), and collagen type II (Col2) staining of elastic cartilage tissue after RWV culture (in vitro), transplanted elastic cartilage tissue (in vivo), and primary cartilage. Scale Bars: 200 μm. (**B**) Shear stress values of cultured elastic cartilage tissue (in vitro), transplanted elastic cartilage tissue (in vivo), and primary cartilage. Holm–Sidak multiple comparisons test: * *p* < 0.01 vs in vivo samples (*n* = 4–41) (**C**) Appearance scale changes over time up to 8 weeks after transplantation. Holm–Sidak multiple comparisons test: * *p* < 0.01 vs. 1 and 2 w after transplantation (*n* = 22–28).

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
