# Peer review of "Development of a Method for Scaffold-Free Elastic Cartilage Creation"

_ijms, 2020, doi:10.3390/ijms21228496_

Round 1

Reviewer 1 Report

This is a fine example of hypothesis-driven science. I have no major concerns although it would be pertinent to know if the authors did any additional extensive analysis of the cartilage-construct extracellular matrix besides the analysis for Type I and II collagens.

Author Response

Reviewer 1

This is a fine example of hypothesis-driven science. I have no major concerns although it would be pertinent to know if the authors did any additional extensive analysis of the cartilage-construct extracellular matrix besides the analysis for Type I and II collagens.

→We would like to thank reviewer 1 for addressing this important comment. In this manuscript we analyzed only Type I and II collagens for ECM detection.

Reviewer 2 Report

The Manuscript “Development of a method for scaffold-free elastic cartilage creation” addresses a really important topic, which deals with finding new therapeutical approaches for Microtia. However, the Authors should adjust and improve the quality of the manuscript before considering it for publications:

The Abstract must be changed, and the Authors should focus on the results they obtained in the study they conducted, and they do not be limited to provide some general information about the conventional used methods.

Please write the whole name of the abbreviated used terms when written for the first time within the text not only within the images.

Line 59: delete “in”.

The Introduction section is too small, and it does not explore the previously done works using scaffold materials and their consecutive limitations to regenerate the auricular cartilage tissue. Please improve the Introduction by citing other works that focus on the regeneration of the of cartilage tissue, the obtained results as well as the limitations.

Figure 1: there are some interrogative marks on DAPI COL1 COL2 image, on Multi Layer Culture (B), and RWV. Please adjust the figure. Please add also the letter C within the Figure 1.

Figure 2: Please adjust the errors occurred in the Figure as well as the other Figures. Remove all the interrogative marks.

The Authors talked about in-vivo results without showing any macroscopical evidence neither the transplanted site before and after transplantation. Please add some images confirming the phrase in the paragraph 2.4. “The regenerated cartilage tissue after transplantation showed shrinkage with 2 weeks, but 129 transplanted cartilage was stable, and there was no significant difference between 1 and 3 months.”

Line 186: Paragraph 4.2. Please specify the number of elastic cartilage samples collected.

Line 225: Paragraph 4.5. Please specify the number of used animals and add a described figure that shows the conducted in vivo experiments.

Line 240: Paragraph 4.7. Please Write in detail the protocols used for Col1 and Col2 staining.

Line 244: Paragraph 4.8. Please Write in detail the ELISA protocol and the sample preparation.

Line 259: Paragraph 4.11. Place this paragraph at the beginning of Materials and Methods Section after 4.1. Animals

Line 265: Paragraph 4.12. Please specify the number of replicates and the number of analyzed samples.

Line 144: The Authors stated “By enhancing the function of cartilage progenitor cells by the layering and rotation culture conducted this time, the cartilage tissue formed a three-dimensional cartilage tissue of 1000 mm3 and succeeded in reproducing the hardness, similar to that of human auricular cartilage. Previous studies have failed to create a thick cartilage tissue.” The Authors did not provide any reference values about the cartilage hardness in human and they did not cite any previous work that confirm the phrase talking about the studies that failed to create a thick cartilage tissue.

Line 158-160: The Authors mentioned that the biodegradability of the engineered scaffold may compromise the regeneration of the tissue caused by the reduction of the reconstructed cartilage. It is very known that the synthetized scaffolds can have different degradation profiles based on their physicochemical properties and the degradation profile can be controlled according to the regeneration period. Please discuss better this issue by providing evidence about the compromised regeneration of auricular cartilage in vivo due to the size reduction of the implanted scaffolds as well as the regeneration failure caused by the scaffold application.

Lines 155-162: The Authors mentioned two terms “regenerated cartilage” and “transplanted cartilage” and they talked about “shrinkage after transplantation during maturation” without providing any technical materials. The authors are invited to provide pictures about the immature and mature cartilage tissue before and after transplantation.

The Authors used the human cartilage progenitor cells to fabricate the 3D cartilage tissue and they transplanted it subcutaneously within mouse models. However, the authors did not address nor discuss the compatibility or graft-rejection/acceptance of the conducted xeno-transplantation. The Authors must better clarify this part and provide some images of the transplanted sites after transplantation. It could be also interested to evaluate the inflammatory response within the transplanted tissue.

Author Response

Reviewer 2
The Manuscript “Development of a method for scaffold-free elastic cartilage creation” addresses a really important topic, which deals with finding new therapeutical approaches for Microtia. However, the Authors should adjust and improve the quality of the manuscript before considering it for publications:

The Abstract must be changed, and the Authors should focus on the results they obtained in the study they conducted, and they do not be limited to provide some general information about the conventional used methods.
→ Thank you for the important comment. According to the reviewer 2`s comment we modified the abstract and added the sentences (page 1 line 30-34).

Please write the whole name of the abbreviated used terms when written for the first time within the text not only within the images.
→ Thank you for the important comment. We checked to write the whole name of the abbreviated used terms when written for the first time within the text not only within the images.

The Introduction section is too small, and it does not explore the previously done works using scaffold materials and their consecutive limitations to regenerate the auricular cartilage tissue. Please improve the Introduction by citing other works that focus on the regeneration of the of cartilage tissue, the obtained results as well as the limitations.
→ Thank you for the important comment. We increased introduction and added references in page 2 line 49 to 51 (References 4-9).

Figure 1: there are some interrogative marks on DAPI COL1 COL2 image, on Multilayer Culture (B), and RWV. Please adjust the figure. Please add also the letter C within the Figure 1.
→ Thank you for the comment. We corrected Figure 1 as the reviewer 2`s comment (page 3).

Figure 2: Please adjust the errors occurred in the Figure as well as the other Figures. Remove all the interrogative marks.
→ Thank you for the comment. We corrected Figure 2 as the reviewer 2`s comment (page 4).

The Authors talked about in-vivo results without showing any macroscopical evidence neither the transplanted site before and after transplantation. Please add some images confirming the phrase in the paragraph 2.4. “The regenerated cartilage tissue after transplantation showed shrinkage with 2 weeks, but 129 transplanted cartilage was stable, and there was no significant difference between 1 and 3 months.”
→ Thank you for the important comment. We added the macroscopical view of the transplantation site before and after the transplantation as supplement 1 (page 10).

Line 186: Paragraph 4.2. Please specify the number of elastic cartilage samples collected.
→ Thank you for the comment. In this experiment we used the material taken from 4 patients. We added the number of the patient in page 8 line 211.
Line 225: Paragraph 4.5. Please specify the number of used animals and add a described figure that shows the conducted in vivo experiments.
→ Thank you for the comment. In this experiment we used 41 mice. We added the number of animals in page 8 line 242.

Line 240: Paragraph 4.7. Please Write in detail the protocols used for Col1 and Col2 staining.
→ Thank you for the comment. We added the detail of the protocol for Col1 and Col2 staining in page 9 line 251 to 255.

Line 244: Paragraph 4.8. Please Write in detail the ELISA protocol and the sample preparation.
→ Thank you for the comment. We added the detail of the protocol for ELISA protocol in page 9 line 257 to 268.

Line 259: Paragraph 4.11. Place this paragraph at the beginning of Materials and Methods Section after 4.1. Animals
→ Thank you for the comment. According to the reviewer`s comment we moved 4.11 to 4.1(page 7, line 187 to 192).

Line 265: Paragraph 4.12. Please specify the number of replicates and the number of analyzed samples.
→ Thank you for the comment. According to the reviewer`s comment we added the number of analyzed samples in the figure legends (page 4 line 96 to 98), (page 5 line 116 to 119), (page 6 line 139).

Line 144: The Authors stated “By enhancing the function of cartilage progenitor cells by the layering and rotation culture conducted this time, the cartilage tissue formed a three-dimensional cartilage tissue of 1000 mm3 and succeeded in reproducing the hardness, similar to that of human auricular cartilage. Previous studies have failed to create a thick cartilage tissue.” The Authors did not provide any reference values about the cartilage hardness in human and they did not cite any previous work that confirm the phrase talking about the studies that failed to create a thick cartilage tissue.
→ Thank you for the important comment. In Figure 4B (page 6) we measured the shear stress of the human auricular cartilage (n=4). The previous studies were referenced 10-14 (page 11 line 337 to 350).
Line 158-160: The Authors mentioned that the biodegradability of the engineered scaffold may compromise the regeneration of the tissue caused by the reduction of the reconstructed cartilage. It is very known that the synthetized scaffolds can have different degradation profiles based on their physicochemical properties and the degradation profile can be controlled according to the regeneration period. Please discuss better this issue by providing evidence about the compromised regeneration of auricular cartilage in vivo due to the size reduction of the implanted scaffolds as well as the regeneration failure caused by the scaffold application.
→ Thank you for the important comment. We added these sentences in the discussion according to the reviewer`s comment “The reconstructed cartilage tissue using a scaffold material causes the reduction due to biodegradation of a scaffold material. [12] In addition, the reconstructed cartilage tissue degraded 2 to 4 years after transplantation by inflammatory reaction. [22, 23]” (in Page 7 line 170 to 172).
Lines 155-162: The Authors mentioned two terms “regenerated cartilage” and “transplanted cartilage” and they talked about “shrinkage after transplantation during maturation” without providing any technical materials. The authors are invited to provide pictures about the immature and mature cartilage tissue before and after transplantation.
→ Thank you for the important comment. According to the reviewer`s comment, we change to the term to cultured elastic cartilage tissue (in vitro) and transplanted elastic cartilage tissue (in vivo). In Figure 4 we indicated the histological findings and the shear stress of these tissues (page 6).

The Authors used the human cartilage progenitor cells to fabricate the 3D cartilage tissue and they transplanted it subcutaneously within mouse models. However, the authors did not address nor discuss the compatibility or graft-rejection/acceptance of the conducted xeno-transplantation. The Authors must better clarify this part and provide some images of the transplanted sites after transplantation. It could be also interested to evaluate the inflammatory response within the transplanted tissue.
→ Thank you for the important comment. In supplement 2 we showed the transplantation site. We observed no tumorigenesis of this elastic cartilage tissue and no inflammatory response within the transplanted tissue (supplement 2 page 11).

Round 2

Reviewer 2 Report

The revisioned Manuscript “Development of a method for scaffold-free elastic cartilage creation” is now better improved and the Authors responses to the reviewer questions in general clarified the critical points. However, additional Minor Revision is required to better address some criticism before considering the Manuscript for publication.

PLEASE…The authors should adjust and control all the images since there are still some errors (interrogative marks) within some figures and the figures are placed more than one time.

Paragraph 2.4. The authors  should change  the term “tumorigenicity” to immunogenicity”. The test performed by the authors, are referred to immunogenic test and not rejection in the graft site is present. I suppose that “tumorigenicity” term is a mistake!

Paragraph 4.8. The authors are invited to write better explain the detail protocol for Col1 and Col2 staining step by step (final concentration of the antibodies, use of the negative control reaction, negative control tissue etc..). The addition of the used secondary antibodies is not sufficient.

Paragraph 4.12. The authors should specify the number of replicates and the number of analyzed samples in the statistical analysis section together within the images and the other section.

The authors in the previous manuscript specified that they used “the Mann–Whitney U test, calculated comparisons between various points were made using one-way analysis of variance” while in this version they changed it to “Holm-Sidak's multiple comparisons”. The authors are invited to explain this change in the statistical analysis test.

Figure 3: The time points within the graphs are 1w, 2w, 3w, while within the legend the days’ annotation was used. Please uniform the culture time points within the graphs, the figure legends and the text.

Author Response

PLEASE…The authors should adjust and control all the images since there are still some errors (interrogative marks) within some figures and the figures are placed more than one time.

→ Thank you for the comment. We carefully checked and corrected errors in the revised manuscript

Paragraph 2.4. The authors should change the term “tumorigenicity” to immunogenicity”. The test performed by the authors, are referred to immunogenic test and not rejection in the graft site is present. I suppose that “tumorigenicity” term is a mistake!

→ Thank you for the valuable comment. We reconsidered and hanged the term from tumorigenicity to immunogenicity according to the reviewer`s comment. We changed the line 147, 152, 168, 309, 313.

Paragraph 4.8. The authors are invited to write better explain the detail protocol for Col1 andCol2 staining step by step (final concentration of the antibodies, use of the negative control reaction, negative control tissue etc..). The addition of the used secondary antibodies is not sufficient.

→ Thank you for the comment. We added the final concentration of first and second antibodies in page 9 line 251 to 256. Also, we counterstained the nuclei by DAPI and added the explanation in page 9 line 255 to 256.

Paragraph 4.12. The authors should specify the number of replicates and the number of analyzed samples in the statistical analysis section together within the images and the other section.

→ Thank you for the comment. In each examination we explained the number, Figure 2; page 4, line 96, 98. Figure 3; page 5, line 116, 118, 119. Figure 4, page 6; page 6, line 138 to 139.  

The authors in the previous manuscript specified that they used “the Mann–Whitney U test, calculated comparisons between various points were made using one-way analysis of variance” while in this version they changed it to “Holm-Sidak's multiple comparisons”. The authors are invited to explain this change in the statistical analysis test.

→ Thank you for the important comment. In figure 2, 3, and 4 we made the Comparisons between other groups were performed for multiple groups and corrected from the Mann Whitney U test to Holm-Sidak`s multiple comparisons.

Figure 3: The time points within the graphs are 1w, 2w, 3w, while within the legend the days’ annotation was used. Please uniform the culture time points within the graphs, the figure legends and the text.

→ Thank you for the comment. According to the reviewer`s comment we modified the figure legend as page 5, line 115.